# Characterization of Charge States in Conducting Organic Nanoparticles by X-ray Photoemission Spectroscopy

**DOI:** 10.3390/ma14082058

**Published:** 2021-04-19

**Authors:** Jordi Fraxedas, Antje Vollmer, Norbert Koch, Dominique de Caro, Kane Jacob, Christophe Faulmann, Lydie Valade

**Affiliations:** 1Catalan Institute of Nanoscience and Nanotechnology (ICN2), CSIC and BIST, Campus UAB, Bellaterra, 08193 Barcelona, Spain; 2Helmholtz Zentrum Berlin Materialien & Energie GmbH BESSY, D-12489 Berlin, Germany; antje.vollmer@helmholtz-berlin.de; 3Institute of Physics, Humboldt University, D-12489 Berlin, Germany; norbert.koch@physik.hu-berlin.de; 4LCC-CNRS, Université de Toulouse, CNRS, UPS, F-31077 Toulouse, France; kane.jacob@lcc-toulouse.fr (K.J.); christophe.faulmann@cemes.fr (C.F.); lydie.valade@lcc-toulouse.fr (L.V.)

**Keywords:** conducting nanoparticles, tetrathiafulvalene, bis(ethilenedithio)tetrathiafulvalene, charge-transfer complexes, mixed-valence materials, X-ray photoemission spectroscopy, synchrotron radiation

## Abstract

The metallic and semiconducting character of a large family of organic materials based on the electron donor molecule tetrathiafulvalene (TTF) is rooted in the partial oxidation (charge transfer or mixed valency) of TTF derivatives leading to partially filled molecular orbital-based electronic bands. The intrinsic structure of such complexes, with segregated donor and acceptor molecular chains or planes, leads to anisotropic electronic properties (quasi one-dimensional or two-dimensional) and morphology (needle-like or platelet-like crystals). Recently, such materials have been synthesized as nanoparticles by intentionally frustrating the intrinsic anisotropic growth. X-ray photoemission spectroscopy (XPS) has emerged as a valuable technique to characterize the transfer of charge due to its ability to discriminate the different chemical environments or electronic configurations manifested by chemical shifts of core level lines in high-resolution spectra. Since the photoemission process is inherently fast (well below the femtosecond time scale), dynamic processes can be efficiently explored. We determine here the fingerprint of partial oxidation on the photoemission lines of nanoparticles of selected TTF-based conductors.

## 1. Introduction

Among the myriad of techniques used for the characterization of nanoparticles (NPs) we will put the emphasis on X-ray photoelectron spectroscopy (XPS), a well-known and well-established surface-science technique that provides relevant information on the electronic structure of materials including the chemical composition, stoichiometry, and chemical state and environment (e.g., formal oxidation state and bonding) [1,2,3,4]. XPS is a photon-in/electron-out technique, where the samples are irradiated with X-rays of a given photon energy *h*ν (monochromatic or non-monochromatic) and the kinetic energy *K* of the photo-emitted electrons is determined dispersively with an electron energy analyzer. Once the energy references (work function φ and Fermi level *E*_F_) are known, the binding energy (*E*_B_) of the electrons can be obtained typically within an error of ±0.1 eV using the well-known Einstein’s equation *h*ν = *E*_B_ + *K* + *φ* [5].

The inelastic mean free path of electrons in solids is small, in the 1 nm range, and it is a function of the kinetic energy so that XPS is intrinsically a surface-sensitive technique. XPS has been traditionally limited to the ultrahigh vacuum (UHV) environment (below 10^−9^ mbar) in order to strongly reduce surface contamination and to allow the photoelectrons to reach the analyzer within several cm (distance sample-entrance cone of the analyzer). However, the use of XPS has been boosted in the last years thanks to the advent of analyzers that can operate in near-ambient conditions (in the mbar regime), thus allowing the studies of surfaces in more realistic experimental conditions by reducing the so-called pressure gap, the several orders of magnitude difference between UHV and typical industrial conditions [6,7,8]. In this case, the sample-entrance cone must be reduced to 1 mm or less. Recent developments have pushed the ultimate pressures limit above 1 bar close to the sample surface while maintaining the main chamber at a few mbar, further reducing the pressure gap [9].

The application of XPS in the study of NPs spans a host of fields such as catalysis [10,11,12], plasmonics [13,14], sensing [15,16,17], Li-batteries [18], nanobiomedicine [19,20], antibacterial and microbial activity [21], microfluidics [22], and environment [23], to mention but a few.

Here, we will focus on XPS studies performed on NPs of two principal representatives of molecular organic materials based on the π-type donor tetrathiafulvalene (TTF) building block [24]: TTF-TCNQ and (BEDT-TTF)_2_I_3_, where TCNQ stands for tetracyanoquinodimethane and BEDT-TTF for bis(ethylenedithio)tetrathiafulvalene. Figure 1 shows the schemes of the TTF-based molecules referred to in this work as well as of TCNQ. TTF-TCNQ is a quasi-one dimensional metal exhibiting a monoclinic crystal structure built up from parallel, segregated chains of donors (TTF) and acceptors (TCNQ) and with a charge transfer of 0.59 electron/molecule [25,26]. Note that both molecules do not contain any metallic element (i.e., TTF = C_6_H_4_S_4_ and TCNQ = C_12_H_4_N_4_). (BEDT-TTF)_2_X, where X stands for a monovalent anion, builds a series of two-dimensional superconductors exhibiting a large number of polymorphs [27]. The highest critical temperature (*T*_c_) has been reported for the κ-phase (*T*_c_ ≈ 13 K) and the β-phase of (BEDT-TTF)_2_I_3_ exhibits *T*_c_ < 8 K. The crystal structure of β-(BEDT-TTF)_2_I_3_ is built from alternate two-dimensional conducting (BEDT-TTF) and insulating (I_3_^−^) sheets. The formal charge transfer is 0.5, where one electron is shared by two BEDT-TTF molecules (mixed valency), leading to neutral and charged molecules.

In general, the physical properties of molecular organic molecules have been studied using high-quality single crystals, which are usually small and delicate, and to a lesser extent, as thin films. In the last decade, innovative chemical synthesis routes have allowed the preparation of NPs with diameters down to ca. 3 nm [28,29,30,31,32].

## 2. Materials and Methods

The details of the synthesis of TTF-TCNQ and (BEDT-TTF)_2_I_3_ NPs are described in [28,31], respectively. Colloidal solutions of TTF−TCNQ were prepared at room temperature by equimolar quantities of TTF, TCNQ, and *n*-octylamine in tetrahydrofuran (THF) and stirring for 1 h. The NP powder was obtained after evaporation of the solvent and intensive washing with pentane. The NPs exhibited diameters in the 10–35 nm range. β-phase (BEDT-TTF)_2_I_3_ NPs were prepared by the chemical oxidation of BEDT-TTF by iodine (I_2_) in the presence of 1-octanamine, *N*-(2-thienylmethylene). Isolated NPs with diameters between 2 and 6 nm were obtained as well as agglomerated 2–6 nm NPs in 20–30 nm aggregates. Figure 2 shows high-resolution transmission electron microscopy (HRTEM) images of isolated β-(BEDT-TTF)_2_I_3_ NPs and aggregates.

The XPS experiments shown here were performed using two different setups: (i) PHOIBOS150 hemispherical electron energy analyzer (SPECS, Berlin, Germany) with a monochromatic X-ray source (1486.6 eV) operated at 300 W in a base pressure in the low 10^−9^ mbar for the NPs and (ii) SES 100 hemispherical analyzer (Scienta, Uppsala, Sweden) at the SurICat endstation (beamline PM4) at BESSY II using monochromatic 35, 270, and 620 eV photons for the BEDT-TTF thin films in the low 10^−10^ mbar [33].

The NP powders were dispersed on conductive carbon tape strips attached to a stainless steel sample holder and, after a gentle compression with a clean glass microscope slide, the excess powder was removed with nitrogen gas flow. Thin films of neutral BEDT-TTF were deposited by sublimating as-received BEDT-TTF powder, after degassing, with a homemade Knudsen cell in UHV while keeping a Au(111) substrate at room temperature. Previously, atomically clean and ordered Au(111) surfaces were obtained by repeated cycles of Ar-ion sputtering and annealing up to 700 K.

## 3. Results and Discussion

Broadly speaking, the use of XPS can be classified into two levels of complexity. In its simpler form, the chemical information is obtained from the position of the most intense peaks, where the photo-generated electrons leave the solid without losing their kinetic energy (elastic scattering). The binding energies of such lines depend on the chemical environment of the selected atoms and reveal the formal oxidation states and bonding by what is referred to as the chemical shift, and from the peak intensities, the approximate at. % (composition and stoichiometry) can be estimated. This represents the most extended use of XPS, although too often, the popularity and accessibility of the technique leads to incorrect interpretations [34].

The second, more complex level, considers the generation of electron-hole pairs and the interaction of the outgoing electrons with collective excitations such as phonons, plasmons, etc. The hole left behind in the solid is screened differently depending on the electronic structure of the materials (metals, semiconductors, etc.), giving rise to finite lifetimes that define the intrinsic widths of the features, quantified by the full-width at half maximum (FWHM). However, the experimental FWHM contains additional information such as the experimental resolution (X-ray source and analyzer) and contributions from inelastic scattering (e.g., electron-phonon and electron-plasmon interactions) as well as from surface and sub-surface defects [35]. The XPS spectra become more complex when chemical elements with d- and f-orbitals are considered. In this case, the different contributing electronic configurations induced by the removal of the electrons, known as multiplet splitting, leads to rather involved spectra (i.e., there is more than one line per orbital).

Photoemission is an intrinsically fast process, in the 10^−15^ s time domain, becoming an ideal technique to explore the dynamics of electronic processes evolving at lower time scales such as charge transfer in molecular organic materials, which is in the ps (10^−12^ s) range [36]. Due to the π-type donor character of the TTF-derivatives, the focus of the analysis will be mainly centered in the electronic structure of sulfur atoms. XPS has been used to determine the amount of partial oxidation, but only for a few TTF-based materials. The small size of available single crystals limits the number of explored materials.

### 3.1. Charge Transfer in Nanoparticles of TTF-TCNQ

Using in situ cleaved TTF-TCNQ single crystals, Sing et al. showed that the XPS S2p core levels exhibited line shapes that cannot be interpreted in terms of the expected 1/2 ratio between the intensities of the S2p_1/2_ and S2p_3/2_ spin-orbit components, as expected from fundamental quantum physics [37]. A least-squares fit using two doublets with the spin-orbit splitting fixed at 1.18 eV and the referred 0.5 branching ratio renders binding energies of 163.8 and 164.8 eV, respectively, for the two S2p_3/2_ components. We thus observe a shift of 1.0 eV between both lines. In addition, the N1s line shows three components at 398.0, 399.5, and 401.4 eV, respectively. Similar results were obtained for (TMTTF)_2_PF_6_ [24], (TMTSF)_2_PF_6_ [37], and (TMTTF)_2_ReO_4_ [38] single crystals, where TMTTF and TMTSF stand for tetramethyltetrathiafulvalene and tetramethyltetraselenafulvalene, respectively, and on thin films of TTF-TCNQ grown by vacuum sublimation for the N1s lines, with binding energies of 398.0 and 399.8 eV, respectively, for the most prominent features [39].

Figure 3 shows the XPS S2p and N1s lines using monochromatic AlKα radiation (1486.6 eV) of a dispersion of NPs of TTF-TCNQ stabilized in the presence of *n*-octylamine. Appendix A depict the survey and C1s spectra, respectively.

A least-squares fit of the S2p spectrum (see Figure 3a) using a fixed spin-orbit splitting of 1.2 eV and the 0.5 branching ratio and imposing equal areas for the combined Gaussian–Lorentzian functions for the main components (represented by continuous blue and red lines) gives three doublets at 163.9 (blue line), 165.0 (red line), and 167.7 eV (olive line), respectively. The first two, and more intense, correspond to those previously reported [37], with a shift of 1.1 eV. The higher energy value (167.7 eV) is characteristic of SO_3_^2−^/SO_4_^2−^ moieties, but we exclude such origin in our case since no sulfur-based chemicals (apart from TTF) were used during the synthesis. We ascribed the 167.7 eV line to surface charging originating from inhomogeneities (e.g., TTF-rich regions surrounding the NPs). This is corroborated by the observed excess of TTF over TCNQ extracted from the surface composition given in Appendix A. A similar effect has been previously observed in as-received measured single crystals of (TMTTF)_2_PF_6_ [24]. In the case of the N1s line (see Figure 3b), two intense features at 398.3 and 399.9 eV, respectively, and a shoulder at 401.6 eV were observed. The higher energy feature arises from a shake-up process between the highest occupied molecular orbital (HOMO) of the neutral and the lowest unoccupied molecular orbital (LUMO) of the ionized TCNQ molecule [40] and from the presence of hydrogen bonded and protonated amine groups as well as of N‒O bonding [41]. The 399.9 eV feature may contain contributions from unprotonated amines as well as from amides resulting from the exposure of the NPs to the ambient. Note the agreement of the binding energy values with those from the single crystal above-mentioned.

We conclude that the 163.9 and 165.0 eV features were ascribed to the neutral and positively charged TTF molecules, respectively, and the 398.3 and 399.9 eV lines to the negatively charged and neutral TCNQ molecules, respectively, and that the charge fluctuations between TTF and TCNQ were detectable with photoemission because the time scales associated to charge transfer are slower than those of the photoemission process itself [37].

### 3.2. Electron Binding Energies of Neutral Molecules

An interesting property shown by some TTF-based molecules such as TMTTF and BEDT-TTF is that both neutral and fully charged molecules are stable. Such property makes them ideal to explore the binding energies of such molecules as a function of the charge state. However, the solids built by such neutral molecules are insulators, which induce surface charging due to the removal of electrons by the photoemission process and thus a non-zero shift of the positions of the core levels toward higher binding energies (lower kinetic energies) can be expected. The determination of the binding energy of the S2p core level of neutral molecules is discussed next.

When neutral TTF molecules are deposited on atomically clean and ordered Au(111) surfaces in UHV in the submonolayer regime, the S2p lines show two doublets at 163.6 and 164.0 eV, respectively [33]. In this case, the shift between both doublets is 0.4 eV, which is lower than that shown by TTF-TCNQ, as discussed above (about 1 eV). Using Au(111) substrates is most indicated since the binding energy of the Au4f_7/2_ line (84.0 eV referred to the Fermi level) can be taken as an absolute reference. Density functional theory (DFT) calculations have shown that a large surface dipole builds at the TTF/Au interface due to the charge transfer from the TTF molecule to the metallic surface [42]. As a result, the TTF molecule becomes positively charged (about +0.6 electron/molecule) and the surface negatively charged (about −0.4 electron/molecule). Thus, from the photoemission point of view, if the charge donation can be considered as a dynamic process, in addition to the component corresponding to the neutral molecule, a higher binding energy contribution corresponding to the charged molecule should appear. The intensity of the 163.6 eV feature increases for increasing coverage, which suggests that the number of neutral molecules increases due to their decoupling from the metallic surface [33].

In order to determine the binding energy of the S2p_3/2_ line for neutral molecules, we deposited BEDT-TTF molecules on both clean Au(111), as discussed above for TTF [33], and on Au(111) covered with C_60_ molecules in order to electronically decouple the BEDT-TTF molecules from the metallic surface. Figure 4a shows XPS S2p spectra taken with 270 eV photons at different coverages of BEDT-TTF on Au(111). We used 270 eV photons in order to enhance the surface sensitivity of the measurement, since for S2p, the resulting kinetic energy is about 100 eV, an energy that shows a minimum of the inelastic mean free path for electrons. From the spectra, at least four features were observed with energies around 161.2, 162.3, 163.4, and 164.5 eV, with the 163.4 eV feature becoming more intense for increasing coverages. The 161.2 eV (S2p_3/2_) feature is characteristic of sulfur-gold bonding, as found for thiol-based molecules self-assembled on Au(111) [43,44,45], while the 162.3 eV component corresponds to the associated S2p_1/2_ line. The BEDT-TTF molecule binds to the surface through its sulfur atoms contained in the ethylenedithia (–S–CH_2_–CH_2_–S–) external groups and the central TTF core should remain essentially unaffected (i.e., neutral), represented by the spin-orbit splitting 163–164 eV features. Thus, in order to avoid the strong sulfur-gold bonding, the BEDT-TTF molecules should be electronically decoupled from the gold surface. The same effect has been observed for the asymmetric ethylenedithia-tetrathiafulvalene (EDT-TTF) molecules grown on Au(111) surfaces (not shown).

The binding energies shown in Figure 4a were calculated using the actual work function for each coverage by applying the photoemission expression mentioned in the Introduction. Figure 4b shows the secondary electron cut-off, also known as the photoemission onset, taken with 35 eV photons as a function of coverage. The cut-off was obtained by applying a −10 V bias to the sample in order to clear the analyzer work function. φ was calculated by subtracting the spectral weight (difference between cut-off and Fermi level positions) from the photon energy. For the clean Au(111) surface, we obtained φ = 5.54 eV, characteristic of crystalline Au(111) surfaces [46]. Note that the work function rapidly decreased already for 0.08 ML (4.98 eV) and saturated above 0.5 ML (4.9 eV). The value corresponding to a thick film (nominally 30 ML), which was grown as a reference, was 4.68 eV. Within the 0.08–6 ML range, the binding energy of the most intense feature was 163.24 ± 0.05 eV. Note that for a thick film, the binding energy was 163.6 eV, an increase that should arise from surface charging due to the insulating character of the film. The binding energy was alternatively determined using the Au4f_7/2_ eV reference, with a binding energy of 84.07 eV as determined for clean and ordered Au(111) single crystals using the SPECS PHOIBOS150 analyzer. From the energy difference between the Au4f_7/2_ and S2p_3/2_ lines measured with 270 eV photons, we obtained 163.28 ± 0.08 eV and 163.5 eV for the thick film. The binding energy values obtained for S2p_3/2_ using both methods are similar, as expected. Using 620 eV photons (see Appendix A) the resulting binding energy for the 0.08–6 ML range was 163.30 ± 0.08 eV and 163.3 eV for the thick film, indicating that the observed higher 163.6 eV value reported using 270 eV photons was certainly due to surface charging.

Figure 4c shows the evolution of the valence band taken with 35 eV photons at normal emission as a function of coverage. As expected, the intensity at *E*_F_ (0 eV) decreased with increasing coverage. For the thick film, *E*_F_ was still visible, indicating the formation of a non-homogeneous film. Note the intensity reduction of the features at about 3 and 4.5 eV observed for clean Au(111) upon deposition of the BEDT-TTF film as well as the observation of the HOMO band building up around the 0.5 eV binding energy, which became more evident for the thicker film.

The electronical decoupling between BEDT-TTF and Au(111) was obtained by covering the Au(111) surface with a thin C_60_ film. Figure 5a shows the XPS S2p lines of a 0.17 (black line) and 16 ML (orange line) films of BEDT-TTF grown on a thick layer (about 5 nm) of C_60_ deposited on Au(111) measured with 270 eV photons. The binding energies were determined from the corresponding work function values (see Appendix A). These layers were deposited subsequently in UHV, without exposing the surfaces to ambient conditions. The figure reveals the presence of only one doublet. A least-square fit using a combination of Gaussian–Lorentzian functions and a Shirley-type background subtraction and imposing the same FWHM for both components and a branching ratio of 0.5 provided a binding energy of the S2p_3/2_ line of 163.9 eV with a spin-orbit splitting of 1.2 eV and a FWHM of 0.68 eV (see Figure 5b). Using 620 eV photons (see Appendix A), the resulting binding energies for both the 0.17 and 16 ML films were 163.6 eV.

Thus, despite the dispersion of values of the binding energy of the S2p_3/2_ line, which spans from 163.3 to 163.9 eV, obtained using different samples (NPs and thin films), photon energies, and calculation methods in this work, we can safely conclude that the origin of the reported 163.8 eV value in [37] arises from the contribution of the neutral TTF core.

### 3.3. Mixed Valency in Nanoparticles of β-(BEDT-TTF)_2_I_3_

Let us now study the case of superconducting β-(BEDT-TTF)_2_I_3_ NPs [31]. Figure 6 shows the XPS S2p (a) and I3d (b) lines obtained with 1486.6 eV photons. Survey and high-resolution spectra from the C1s line are presented in Appendix A.

The least-square fit shown in Figure 6a shows a dominant doublet with a S2p_3/2_ line centered at 163.7 eV and a residual contribution at 164.8 eV. Analogous studies performed on single crystals of BEDT-TTF salts provide binding energies of 164.0 eV (low-resolution) [47] and 163.9 eV (high-resolution) [48]. Note that the contribution associated with the charged BEDT-TTF state is marginal, as if charge transfer was not activated, a fact that is in conflict with the observed superconducting transition and with the used synthetic route, where BEDT-TTF is chemically oxidized by iodine (I_2_) [31].

The analysis of the XPS lines of the I3d core levels, shown in Figure 6b, can provide some clues on this contradiction. The spectrum reveals a large spin-orbit splitting (11.46 eV) between the 5/2 and 3/2 components. Each component exhibits a finer structure with two dominant and three minor contributions, respectively, which can be observed from the least-squares fit of the I3d_5/2_ line. The observed structure responds to the differential contributions from the central (619.8 eV) and two side atoms (618.6 eV) in the linear I_3_^‒^ anion, represented by continuous red and blue lines, respectively [49,50]. The features centered at 623.5 eV are due to a shake-up satellite associated to the side hole [49]. The weak feature at 621.0 eV, represented by a continuous olive line could be ascribed to contributions from I_5_^−^ anions [51]. The chemical shift associated with the main lines amounts to 1.2 eV and their intensity ratio was 0.65, exceeding the expected 0.5 value (two side atoms per one central atom in the molecule). This difference can be explained by the relaxation of symmetry constraints, which affect only the side atoms in the linear molecule [49]. Thus, during the synthetic route, iodine is effectively reduced to the tri-iodine anion, according to the reaction 3I_2_ + 2e^−^ → 2I_3_^−^.

The small contribution of the charge BEDT-TTF species to the XPS spectra shown in Figure 6 should not be associated with surface alterations caused by the X-ray beams (beam damage) since this would also affect the tri-iodine anion and an extra signal at about 620 eV should be expected, which would modify the observed ratio [51]. The reduced number of XPS experiments performed on BEDT-TTF salts makes it difficult to draw conclusions, so additional investigations are needed. With our results, we can hypothesize the contribution of the surface effects, where neutral BEDT-TTF molecules would be concentrated at the surface of the NPs or dispersed among NPs within the aggregates. This is in line with previous studies of (TMTSF)_2_ClO_4_ NPs, where the superconductivity transition has been verified with an onset at about 1.2 K, the same transition temperature observed for single crystals [52]. XPS of the Se3d line of such (TMTSF)_2_ClO_4_ NPs revealed several components, with a dominant Se 3d_5/2_ feature at 55.7 eV, which corresponds to TMTSF in the neutral state, and less intense contributions at 56.1, 57.0, and 58.8 eV, respectively. Such features might be assigned to different oxidation states of TMTSF, thus evidencing different degrees of charge transfer, although one would expect only the contribution from formal TMTSF^+0.5^ [30].

## 4. Conclusions

The well-known and accessible XPS technique is an excellent tool to explore charge donation in conducting molecular materials derived from the TTF molecule due to the shorter time scales found in the photoemission process compared to those associated with charge transfer. This means that neutral and charged molecules can be discriminated with XPS. The binding energy of the S2p_3/2_ line associated with the neutral molecules was in the 163.3–163.9 eV range, as has been determined on single crystals, nanoparticles, and on thin films of neutral molecules on gold surfaces, while the binding energy of the charged molecules increased with increasing charge donation. 

For TTF-TCNQ NPs, both neutral and charged states were clearly distinguishable but extra contributions appeared at higher binding energies that we associated with surface charging. Such charging may arise from inhomogeneities such as non-stoichiometry at the surfaces of the NPs and matrix effects upon aggregation of NPs. This effect is more dramatic for our β-(BEDT-TTF)_2_I_3_ NPs where the XPS spectra is dominated by the contribution of neutral BEDT-TTF molecules in spite of the demonstrated superconducting behavior of the same powder. An analogous result has been observed for superconducting NPs of (TMTSF)_2_ClO_4_, where different oxidation states have been identified. At this point, we can conclude that the surface sensitivity of XPS accentuates the inhomogeneous character of the isolated and aggregated NPs, a property that does not prevent NPs from exhibiting the same properties as the single crystal counterparts. Within this framework, we encourage the research in organic NPs and their exploration with XPS.

## Figures and Tables

**Figure 1 materials-14-02058-f001:**
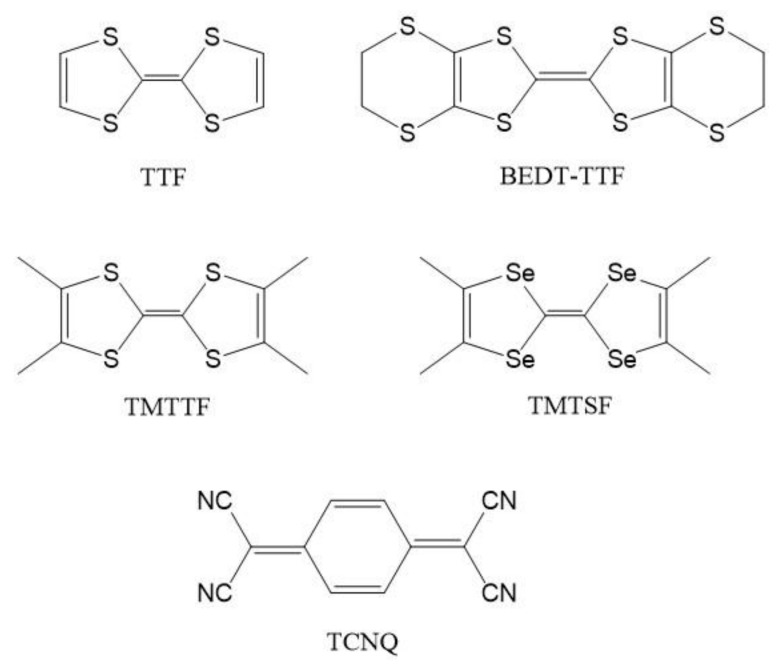
Molecular structures of TTF (tetrathiafulvalene), BEDT-TTF (bis(ethylenedithio)tetrathiafulvalene), TMTTF (tetramethyltetrathiafulvalene), TMTSF (tetramethyltetraselenafulvalene), and TCNQ (tetracyano-quinodimethane).

**Figure 2 materials-14-02058-f002:**
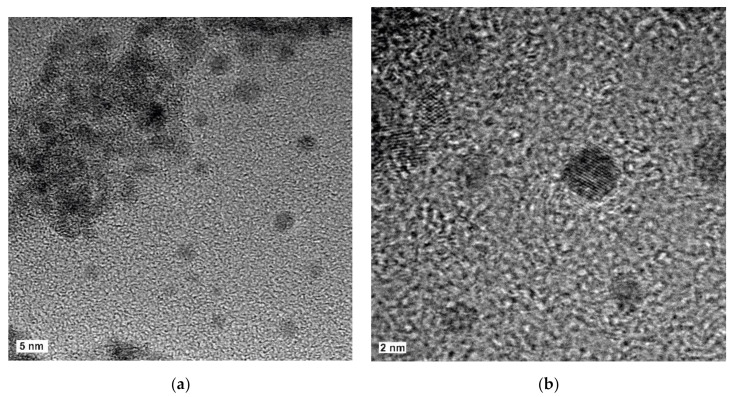
HRTEM images of aggregated and isolated nanoparticles of β-(BEDT-TTF)_2_I_3_ (**a**) and of isolated nanoparticles showing crystallographic planes (**b**). The images were recorded with a FEI Tecnai F20 HRTEM operating at 200 kV. The samples were sonicated, dispersed in acetonitrile, and placed onto a holey carbon-copper support grid.

**Figure 3 materials-14-02058-f003:**
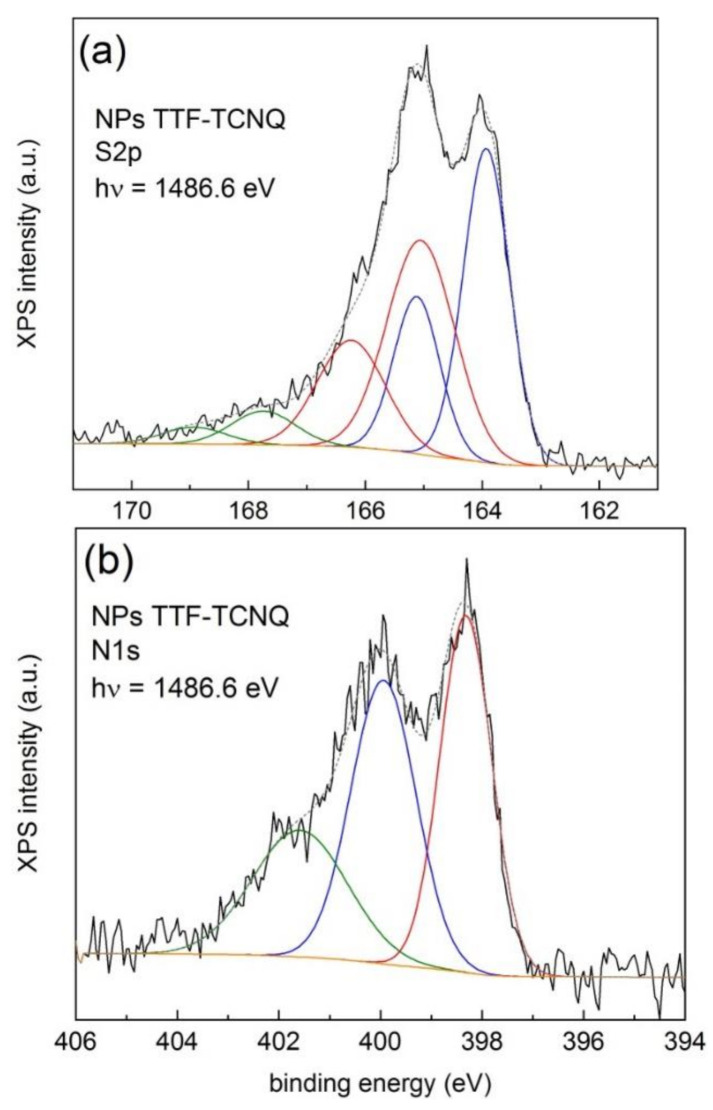
High-resolution XPS S2p (**a**) and N1s (**b**) lines of a dispersion of NPs of TTF-TCNQ stabilized with *n*-octylamine.

**Figure 4 materials-14-02058-f004:**
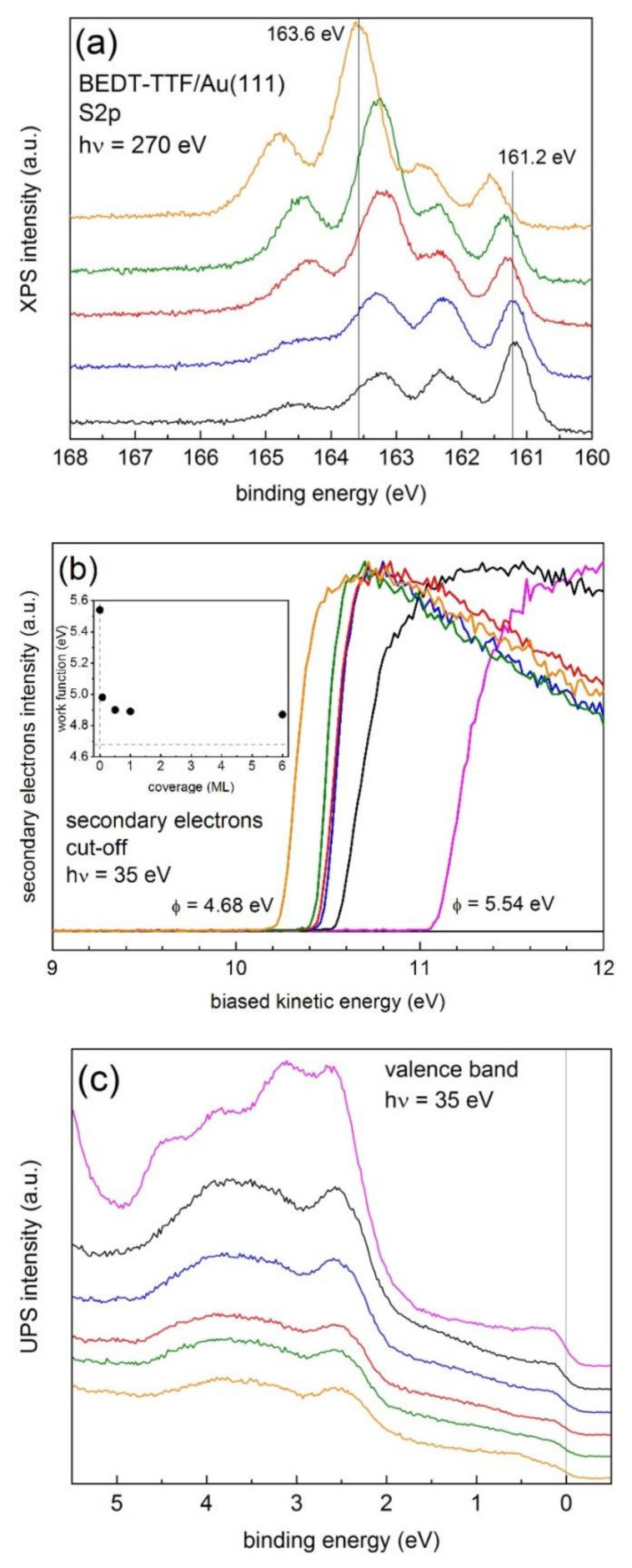
(**a**) High-resolution XPS S2p lines of BEDT-TTF/Au(111) as a function of coverage obtained with 270 eV photons. (**b**) Photoemission onset (secondary electron cut-off) as a function of coverage obtained with 35 eV photons. The inset shows the calculated work function as a function of coverage. The value corresponding to the thick BEDT-TTF film (4.68 eV) is represented by a discontinuous horizontal line. (**c**) Valence band spectra as a function of coverage. Color codes: clean Au(111) surface (magenta line), 0.08 ML (black line), 0.5 ML (blue line), 1 ML (red line), 6 ML (green line), and a thick BEDT-TTF film (orange line).

**Figure 5 materials-14-02058-f005:**
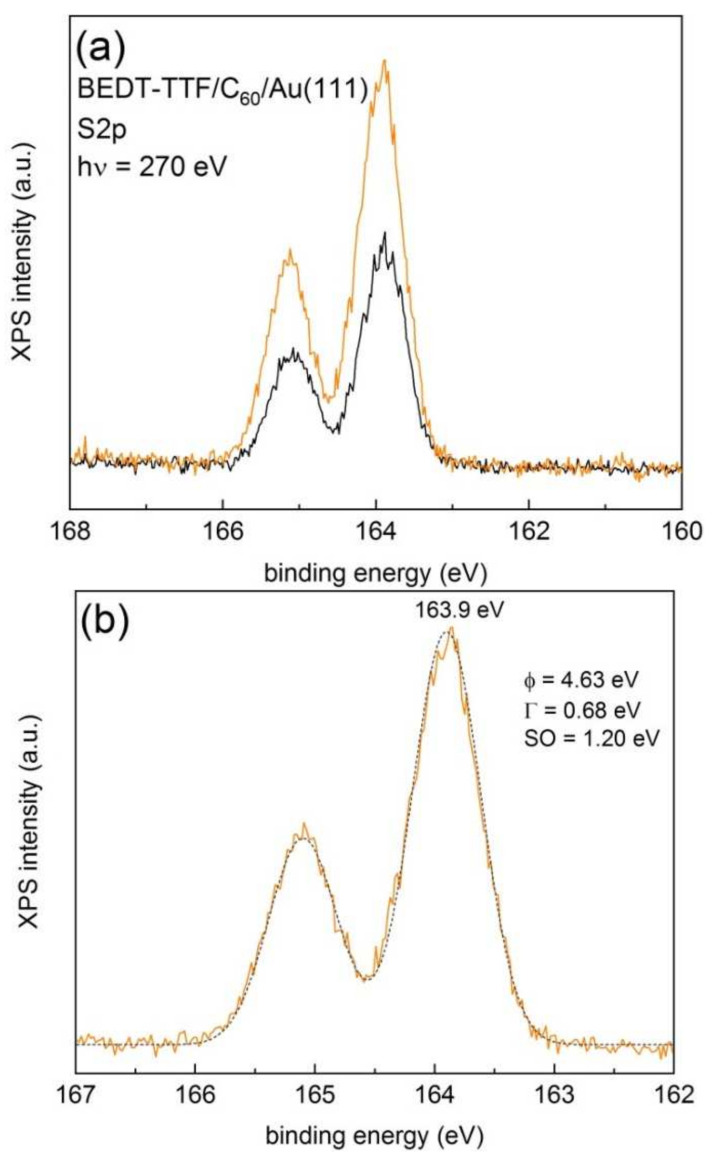
(**a**) High-resolution XPS S2p lines of a 0.17 ML (black continuous line) and 16 ML (orange continuous line) films of BEDT-TTF grown on a 5 nm thick layer of C_60_ deposited on Au(111). (**b**) Least-square fit of the 16 ML film using a combination of Gaussian–Lorentzian functions and a Shirley-type background subtraction and imposing the same FWHM for both components and a branching ratio of 0.5. The discontinuous grey line represents the envelope of the fit.

**Figure 6 materials-14-02058-f006:**
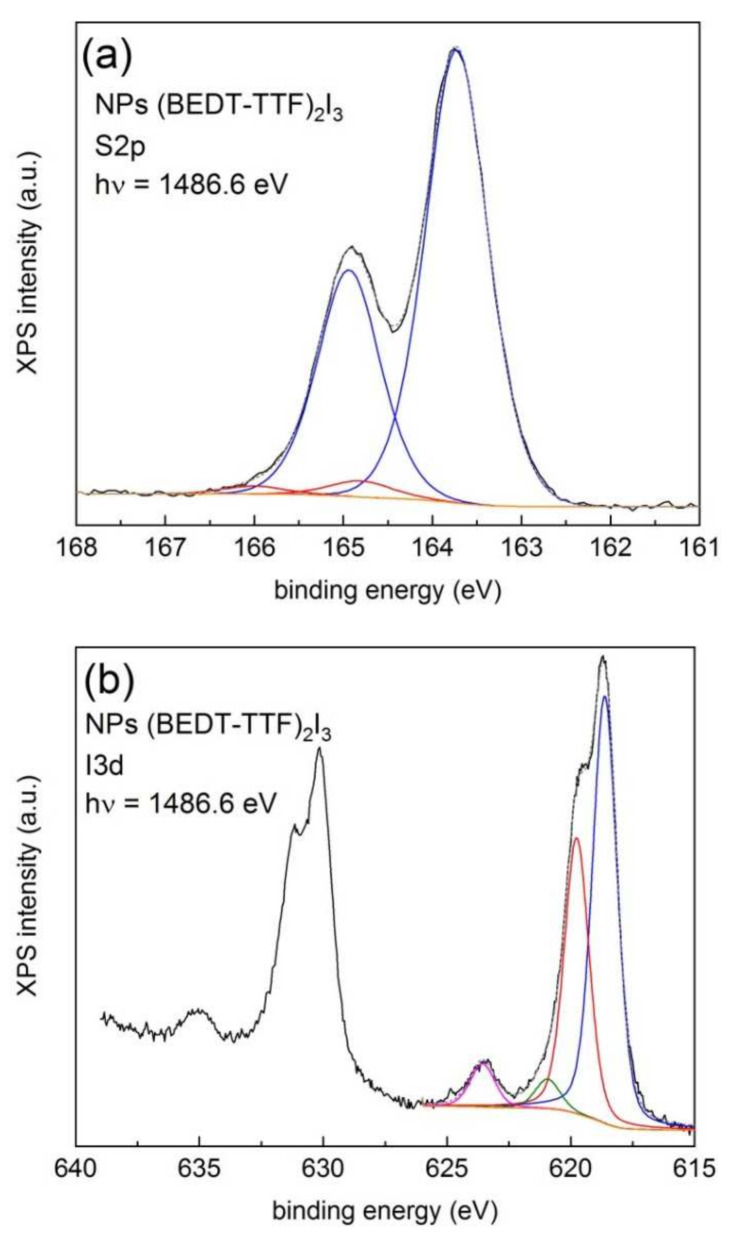
High-resolution XPS spectra of the (**a**) S2p and (**b**) I3d lines of (BEDT-TTF)_2_I_3_ NPs.

## Data Availability

The authors confirm that the data supporting the findings of this study are available within the article.

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
