# Peer review of "Characterization of Charge States in Conducting Organic Nanoparticles by X-ray Photoemission Spectroscopy"

_materials, 2021, doi:10.3390/ma14082058_

Round 1
Reviewer 1 Report
The manuscript “Characterization of charge states in conducting organic nano-2 particles by X-ray photoemission spectroscopy” report the XPS characterization of different organic compounds with particular focus on the charge transfer states. Despite its potential interest for the readers of Materials, I think the manuscript must be improved under different aspects and most data must be provided before being accepted for publication.
- Improve the quality of Figure 1, at least in my version of the text, it is very grainy and not well defined.
- In Figure 2 please increase the size of the white scale bar (bottom-left corner), it is not clearly visible
- Line 98 “The XPS experiments shown here were performed using two different setups: …”
while you explain that you used the SPECS PHOIBOS150 hemispherical analyser with Al Ka radiation to study NP, there is not explicit description of the samples measured at the synchrotron. Please add for clarity. - Line 128 “ This is due to the fact …detectable”
I don’t see how the ultrafast nature of the photoemission process is involved in the detection of multiplet splitting due to spin-orbit coupling. Could you please clarify? I think this phrase would be better referred to “long life time” excitations like plasmons shake up etc. and also in this case may be misleading. - In all the figures depicting XPS spectra, you insert the numerical value of the photon energy but it is rather confusing in this way. Please add hv= in front of this numerical value in order to make immediately clear that is a photon energy, or remove the number and write the photon energy in the caption of the figure.
- In section 3.2, you study BEDT-TTF/Au(111) as a function of film thickness. However film thickness and coverage are often used as the same word. Please re-read this section and pick one of the two to use homogeneously. For example in figure 4a: ”High-resolution XPS S2p lines of BEDT-TTF/Au(111) as a function of coverage obtained with 270 eV photons…” in the inset of Fig 4a a thickness expressed in nm is reported, not a coverage. I suggest to express the coverage as fraction (or multiples) of monolayer, since for example anything thick 0.0015 nm= 1.5pm seems not reasonable.
Also in figure 4b the x axis of the inset should be modified. - C60 is semiconductor with roughly 2eV of bandgap, how did you manage to deal with the space charge effect?
- Line 227-237
in this paragraph you describe the WF measurements of the sample as a function of coverage with the aim of correctly determine the BE of photoemission peak. I have some doubts here.
- Why do you need the sample WF to determine the BE? In a real photoemission experiments only the analyser work function enters the conservation of energy, it is well known. Because of this the BE is usually referred to Fermi Level instead of vacuum level, and since the analyser and the sample are in electrical contact, it is basically always possible to know the position of FL (after a quick calibration with a metal specimen), even when the sample is insulating or semiconductor. Please clarify this point.
-Why don’t you show the valence band spectra close to FL, but only the secondary electron onset? I think it will increase the quality of the manuscript. - What is the substrate of the NP described in section 3.3? Please insert in the text.
- A legend listing the coverage of the spectra depicted in figure 5a-c is missing.
- At what temperature were performed the XPS measurements? You say “superconducting β-(BEDT-TTF)2I3 NPs”, but were the XPS measurements acquired in the superconductive or normal state? How the superconductive transition may change the charge transfer phenomenon? Please clarify this in the text.
- Why didn’t you measure the NPs Work function? There is a vast literature on the NP size effects over the WF for metals and metals oxide (some example: https://doi.org/10.1016/j.cplett.2011.11.045 ; https://doi.org/10.1021/acs.jpclett.5b01197 ; https://doi.org/10.1039/D0CP00216J) but on the other hand, the literature on organic nanoparticle WF is very scarce. I think comparing the WF of the NP film to a “Regular” molecular thin film will increase dramatically the value of the paper.
- In the Title you say “… conducting organic nanoparticles” however, there are no proofs of their conductivity. The manuscript interest will increase if transport measurement would be added to the text, or at least some VB spectra would be shown. It would be also interesting to compare the transport behaviour of these nanoparticles and a thin film.
- Any kind of structural characterization of the samples is missing. Could you please add some Raman spectra of your films? Quantum size effects may be also detectable on the vibrational mode
Reviewer 2 Report
The presented manuscript contains a well-organized experiment from a technical point of view. The results are presented clearly, the analysis is consistent with the findings, and the conclusions are beyond doubt. At the same time, the work looks rather weak from the point of view of the novelty and importance of the results obtained. I think authors should pay attention to this point when submitting a revised manuscript.
Nevertheless, I believe that the work can be accepted taking into account the comments below.
1) In the introduction, the authors point out the possibility of determining the position of the line with an accuracy of 0.1 eV, which is less than the instrumental resolution of most modern laboratory spectrometers. I think this statement looks somewhat crude, although it can be accepted for conducting samples.
2) In Fig. 2 very small labels on the size scales.
3) Despite the focus of the article on S 2p levels, I believe that Survey spectra as well as C 1s are worth showing and briefly discussing. At least provide it for reference in supplementary materials.
4) Fig. 3а. Could the 167.7 eV peak be part of the background? If, for example, we take into account the background according to the Tougaard method?
5) I would like to know the amount of impurity oxygen (you can also give the surface compositions in supplementary materials) in order to take this into account when interpreting the 401.6 eV band in Fig. 3b. It is also not clear why the authors exclude the possibility of the formation of N-H bonds, which may be responsible for the 399.9 eV band.
6) It is not clear from the work whether the authors themselves measured the Se 3d spectra. If so, they also need to be shown. If this point concerns only the analysis of the literature, then it is worth revising the conclusions.
Round 2
Reviewer 1 Report
The authors replied convincingly to all my questions. I recommend this article for publication.